# UniPose: Detecting Any Keypoints

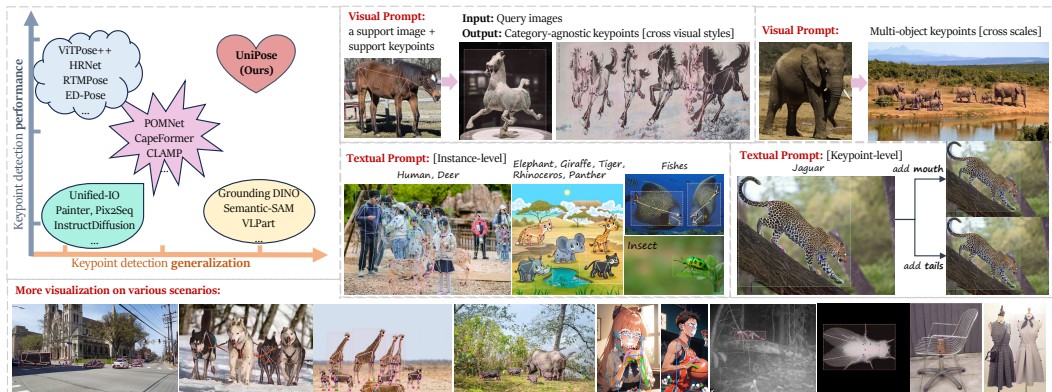

Figure 1: The proposed *UniPose* achieves both high keypoint detection generalization ability and high performance. *UniPose* utilizes visual and textual prompts for training to learn fine-grained local-region visual representation via keypoint-text and keypoint-image alignments. Once trained, it can generalize cross-instance and cross-keypoint categories, where it can detect multi-object keypoints on various challenging scenarios with diverse visual styles, scales, and poses.

## Abstract

This work proposes a unified framework called *UniPose* to detect keypoints of any articulated (e.g., human and animal), rigid, and soft objects via visual or textual prompts for fine-grained vision understanding and manipulation. Keypoint is a structure-aware, pixel-level, and compact representation of any object, especially articulated objects. Existing fine-grained promptable tasks mainly focus on object instance detection and segmentation but often fail to identify fine-grained granularity and structured information of image and instance, such as eyes, leg, paw, etc. Meanwhile, prompt-based keypoint detection is still under-explored. To bridge the gap, we make the first attempt to develop an end-to-end prompt-based keypoint detection framework called *UniPose* to detect keypoints of any objects. As keypoint detection tasks are unified in this framework, we can leverage 13 keypoint detection datasets with 338 keypoints across 1,237 categories over 400K instances to train a generic keypoint detection model. *UniPose* can effectively align text-to-keypoint and image-to-keypoint due to the mutual enhancement of textual and visual prompts based on the cross-modality contrastive learning optimization objectives. Our experimental results show that *UniPose* has strong fine-grained localization and generalization abilities across image styles, categories, and poses. Based on *UniPose* as a generalist keypoint detector, we hope it could serve fine-grained visual perception, understanding, and generation.

## 1 Introduction

Keypoint detection is a fundamental computer vision task that estimates the 2D keypoint positions of any object in an image. It is of great impact to robot and automation, VR/AR, neuroscience, biomedicine, and human-computer interaction areas. Keypoint can describe compact structure information at the pixel level, thus representing fine-grained and local visual information which is very helpful for behavioral analysis and performing manipulation (*e.g.*, animating the object). Specifically, due to the increasing real-life application needs, 2D human pose estimation plays an important

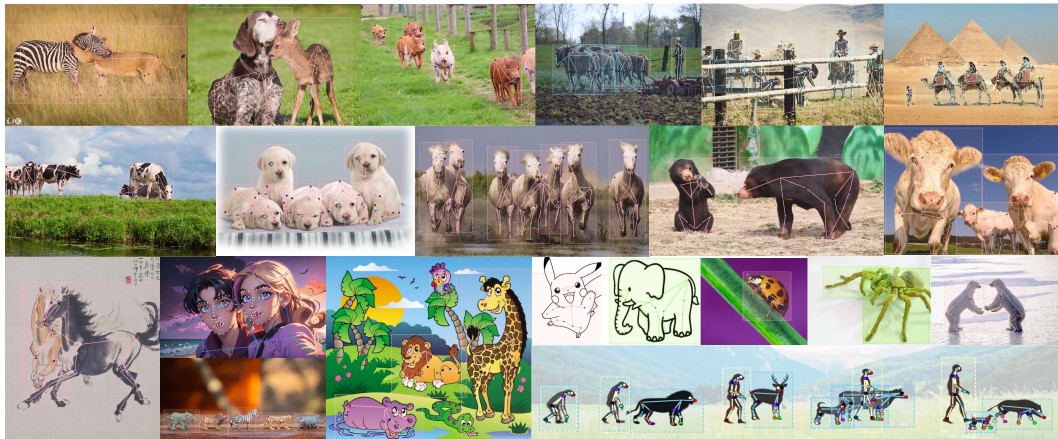

Figure 2: Qualitative results of the proposed *UniPose* on arbitrary in-the-wild images. We highlight the powerful detection performance from cross-category (the first row), multi-object (the second row), and cross-image-style (the third row) with various pose scenarios.

role in this area, which focuses on detecting multi-person keypoint (e.g., head, hand, and foot keypoints) (Xu et al., 2022b; Cheng et al., 2020; Jiang et al., 2023; Yang et al., 2022a; 2023b). To study animal behaviors in zoology and wildlife conservation, some works propose to perform animal pose estimation (Yu et al., 2021; Sun et al., 2023a; Ye et al., 2022; Mathis et al., 2018; Xu et al., 2023; Zhang et al., 2023). However, these studies can only detect object keypoints of a single-class. Imagine if we need to analyze the behavior of various species of animals and human interactions; existing solutions need to train many category-specific models for different species.

Although arbitrary object detection and segmentation has made great progress (Kirillov et al., 2023; Liu et al., 2023b; Sun et al., 2023b; Liang et al., 2023; Zhong et al., 2022), there are few explorations on the problem of multi-object keypoint detection of unseen or arbitrary categories. The problem is non-trivial because it needs to learn fine-grained visual representation, category-agnostic keypoint concepts, and semantic structure information. Naively transferring one type of keypoints to another, especially for articulated and deformable objects, is very challenging due to high variations in pose, scale, appearance, background, complicated occlusion, and semantic gaps. Xu et al. (2022a) first proposed the task of category-agnostic pose estimation (CAPE) with visual prompts (*i.e.*, a support image of a novel class and the corresponding keypoint annotations) to estimate the pose of the same class in query images. It formulates it as a keypoint matching problem.

However, existing CAPE methods (Xu et al., 2022a; Shi et al., 2023) have several limitations: 1) only visual prompts are supported, making user interaction unfriendly and inefficient; 2) the keypoint-to-keypoint matching schemes without instance-to-instance matching are not effective and robust since they tend to learn low-level local appearance transformation which often results in inevitable semantic ambiguity without capturing global relations; 3) they use a top-down two-stage detection scheme (*i.e.*, crop the image or use ground-truth boxes for each instance), lacking instance-level generalization ability for handling multi-object scenarios; and 4) the amount of data used for training is usually of small scale (*e.g.*, only 20K images with 100 instance classes), which severely limits the generalizability and effectiveness of the visual prompt-based keypoint detection.

In contrast, human intelligence learns multi-modality information simultaneously and excels at summarizing information through contrastive learning of similarities among categories at different semantic levels. On the one hand, keypoints share similar structures and hold similar appearances cross-species. For instance, as species evolve, skeletal topology is consistent in most quadrupedal mammals, and the eyes of different organisms have similar visual components. On the other hand, visual prompts can only provide pixel-level localization and structure but lack semantic concepts (category-agnostic) from natural language, such as directions (*e.g.*, left, medium, or right), keypoint semantic descriptions (*e.g.*, left eyes of a panda or right collar of a T-shirt). A proper use of text prompts is highly desired to address such deficiencies, and the two kinds of prompts will mutually benefit to image-to-keypoint reasoning and text-to-keypoint alignment.

Considering the above challenges and motivations, we propose to unify keypoint detection tasks in an end-to-end prompt-based framework named *UniPose*, which supports multi-object keypoint detection for unseen objects and keypoints. **First**, we introduce text prompts in the category-agnostic

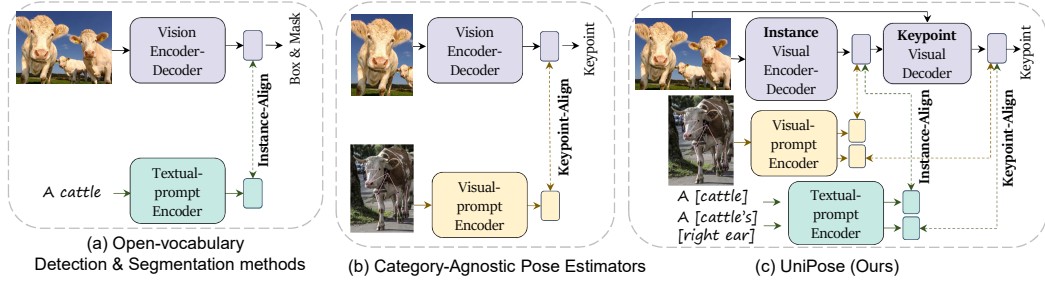

Figure 3: Comparisons of three tasks from the supported inputs and framework overview. *UniPose* utilizes either visual or textual prompts to make any instance-to-keypoint detection effective.

pose estimation task to bring in semantic guidance and relieve the visual ambiguity from existing visual prompts. Through the joint training of both visual and textual prompts in *UniPose*, the semantic understanding and localization capability are reinforced from each other to improve the model's robustness and performance. **Second**, based on the DETR-like end-to-end non-promptable human pose estimator (ED-Pose (Yang et al., 2022a)), we first decode the instance information and then decode the corresponding fine-grained keypoints to provide a coarse-to-fine information flow end-to-end. Moreover, we improve the keypoint-to-keypoint matching strategy into a coarse-to-fine (from image to instance to keypoint) similarity learning process via two kinds of contrastive losses to support multi-object and multi-keypoint detection. **Lastly**, as the quality and quantity of data are both important for effective model training, we unify 13 keypoint detection datasets into 338 keypoints across 1,237 categories over 400K instances by reorganizing inconsistent and undefined keypoints from different datasets and merging similar keypoints and categories. We balance these datasets by considering image appearance and style diversity, instances with varying poses, viewpoints, visibilities, and scales. Each keypoint has its textual prompts, and each category has its default structured keypoint sets. We call the unified dataset *UniKPT*.

Through comprehensive experiments, we show the remarkable generalization capabilities of *UniPose* for unseen object and keypoint detection, which exhibits a notable 42.8% improvement in PCK performance when compared to the state-of-the-art CAPE method. Moreover, *UniPose* outperforms the state-of-the-art end-to-end model (e.g., ED-Pose) across 12 diverse datasets. Its performance is also comparable with state-of-the-art expert models for object detection (e.g., GroundingDINO) and keypoint detection (e.g., ViTPose++). In addition, *UniPose* exhibits impressive text-to-image similarity at both instance and keypoint levels, notably surpassing CLIP by 204% when distinguishing between different animal categories and by 166% when discerning various image styles. As in Fig. 2, we showcase the powerful detection performance of *UniPose* on in-the-wild images and hope it could serve the community for fine-grained visual perception, understanding, and generation.

**Related Work.** Due to the page limit, we present the details in the Appendix A. There are three related areas, including category-specific keypoint detection (e.g., human, animal, cloth pose estimation (Sun et al., 2023a; Ye et al., 2022; Mathis et al., 2018; Ng et al., 2022; Xu et al., 2022b;c; Jiang et al., 2023; Yang et al., 2022a)), category-agnostic keypoint detection (relies on visual prompts) (Xu et al., 2022a; Shi et al., 2023), Open-vocabulary Vision Models (utilizes textual prompts) (Zang et al., 2022; Gu et al., 2021; Li et al., 2022; Yao et al., 2022; Liu et al., 2023b; Liang et al., 2023; Zhong et al., 2022)RegionCLIP (Zhong et al., 2022; Li et al., 2023; Sun et al., 2023b). We show existing prompt-based methodologies in Fig. 3.

## 2 METHOD

*UniPose* is an end-to-end prompt-based keypoint detection framework. It takes an image as input and first decodes instance-level representations (*i.e.*, object bounding boxes), then decodes pixel-level representations (*i.e.*, object keypoints). *UniPose* introduces novel encoding mechanisms for various modalities of prompts and incorporates a novel interaction scheme between the input image and prompts, enabling prompt-based keypoint detection for any object with any keypoint definitions.

**Encoding Multi-modality Inputs.** The input of *UniPose* is a target image to be predicted **I** and the associated user prompts. We offer support for user prompts in two formats: textual descriptions comprising instance or keypoints $\mathbf{P}^t$, as well as instance image $\mathbf{P}^i$ together with its respective

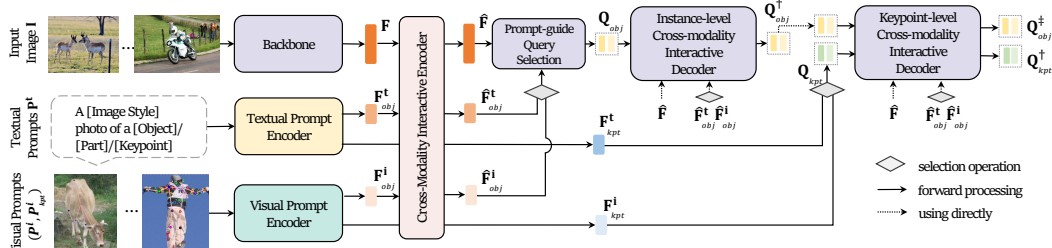

Figure 4: The overview architecture of *UniPose*. Given an input image, *UniPose* follows the coarse-to-fine strategy to detect keypoints of any object via textual or visual prompts.

2D keypoint positions $\mathbf{P}^{\mathrm{i}}_{kpt}$. We employ three distinct modules to encode the corresponding inputs. First, we employ a backbone network to extract multi-scale features of $\mathbf{I}$ and obtain tokenized representations $\mathbf{F}$. Then a Textual Prompt Encoder is adopted to encode $\mathbf{P}^{\mathrm{t}}$ to textual semantic representations $\mathbf{F}^{\mathrm{t}}$, which includes $\mathbf{F}^{\mathrm{t}}_{obj}$ for objects and $\mathbf{F}^{\mathrm{t}}_{kpt}$ for keypoints. At last, we use a Visual Prompt Encoder to encode $\mathbf{P}^{\mathrm{i}}$ and $\mathbf{P}^{\mathrm{i}}_{kpt}$ to visual semantic representations $\mathbf{F}^{\mathrm{i}}$, where $\mathbf{F}^{\mathrm{i}}_{obj}$ and $\mathbf{F}^{\mathrm{i}}_{kpt}$ correspond to objects and keypoints, respectively. The details for prompts encoding are in Sec. 2.1.

**Coarse-to-Fine Keypoint Detection.** Given the representations $\mathbf{F}$, $\mathbf{F}^{\mathrm{t}}$, and $\mathbf{F}^{\mathrm{i}}$, we introduce a Multi-Modality Interactive Encoder to realize interactions among different modalities through cross-attention operations, obtaining the enhanced representations $\widehat{\mathbf{F}}$, $\widehat{\mathbf{F}}^{\mathrm{t}}$ and $\widehat{\mathbf{F}}^{\mathrm{i}}$, respectively. Additionally, we adopt a coarse-to-fine scheme and integrate two decoders that concentrate on different granularities, namely, the Instance-level Cross-Modality Decoder and the Keypoint-level Cross-Modality Decoder. Initially, the prompt-guided query selection is introduced to extract object queries $\mathbf{Q}_{obj}$ from $\widehat{\mathbf{F}}$, which is highly associated with the enhanced object-level semantic representations $\widehat{\mathbf{F}}^{\mathrm{t}}_{obj}$ or $\widehat{\mathbf{F}}^{\mathrm{i}}_{obj}$. Subsequently, the Instance-level Cross-Modality Decoder updates these object queries from $\mathbf{Q}_{obj}$ to $\mathbf{Q}^{\dagger}_{obj}$. The keypoint queries $\mathbf{Q}_{kpt}$ are directly initialized by using $\widehat{\mathbf{F}}^{\mathrm{t}}_{kpt}$ or $\widehat{\mathbf{F}}^{\mathrm{i}}_{kpt}$. We further adopt the Keypoint-level Cross-Modality Decoder to refine both $\mathbf{Q}_{kpt}$ and $\mathbf{Q}^{\dagger}_{obj}$, resulting in $\mathbf{Q}^{\dagger}_{kpt}$ and $\mathbf{Q}^{\ddagger}_{obj}$. The details of the above operations are in Sec. 2.2. Finally, we utilize a Feed-Forward Network to regress keypoint positions with $\mathbf{Q}^{\dagger}_{kpt}$ and object bounding boxes with $\mathbf{Q}^{\ddagger}_{obj}$. Moreover, we employ prompt-guided classifiers for keypoint category classification using $\mathbf{Q}^{\dagger}_{kpt}$ and for object category classification using $\mathbf{Q}^{\ddagger}_{obj}$ (see Sec. 2.3).

## 2.1 MULTI-MODALITY PROMPTS ENCODING

The CLIP model (Radford et al., 2021) is trained on hundreds of millions of image-text pairs, aligning images with their corresponding captions. In this context, *UniPose* leverages its pretrained image encoder and text encoder to encode user prompts through carefully designed encoding mechanisms.

**Textual Prompt Encoder.** 1) *Hierarchical Textual Structure.* To accomplish precise mapping from text to image/region/keypoint, we devise a hierarchical textual structure to describe instance and keypoint, *i.e.* image→instance→part→keypoint. Consequently, we formulate the template as *"A* [IMAGE STYLE] *photo of a* [OBJECT]*"* for the entire instance, *"A* [IMAGE STYLE] *photo of an* [OBJECT]*'s* [PART]*"* for part instances (*e.g.,* face and hand), and *"A* [IMAGE STYLE] *photo of a* [OBJECT]*'s* [PART]*'s* [KEYPOINT]*"* for keypoints. 2) *Textual Prompt Dropout.* Utilizing a hierarchical textual structure equips *UniPose* with specialized retrieval capabilities, such as referring to a particular object category with a specific keypoint. Furthermore, during training, we introduce random dropout for descriptions, including image style, object, or part, to boost its general retrieval capabilities. For instance, hiding the object category promotes the retrieval capabilities of a specific keypoint across all object categories. A typical example is "the left eye of any object".

**Visual Prompt Encoder.** *UniPose* could receive a prompt instance image $\mathbf{P}^{\mathrm{i}}$ along with its corresponding keypoint definitions $\mathbf{P}^{\mathrm{i}}_{kpt}$ (e.g., 2D positions). Its Visual Prompt Encoder aims to encode these prompts into the respective instance and keypoint representations. However, the original CLIP's image encoder (e.g., ViT) can only obtain image representations through the learnable [CLS] token and patch tokens, which are the inputs on the left of Fig. 5-(a). *UniPose* extends this by further incorporating keypoint position encodings, represented as the input on the right of Fig. 5-(a).

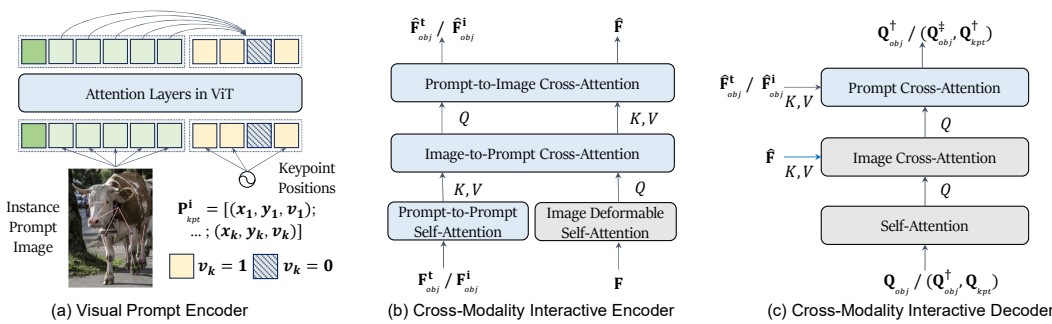

Figure 5: The detailed illustration of (a) Visual prompt Encoder, (b) Cross-Modality Interactive Encoder, and (c) Cross-Modality Interactive Decoder. In (b) and (c), the modules in grey are presented in previous work, while the modules in blue are introduced to incorporate prompt interactions.

1) ***Initialization of Keypoint Tokens.*** Let $\mathbf{P}^i_{kpt} = [(x_1, y_1, v_1), ..., (x_k, y_k, v_k)]$, where $(x_k, y_k)$ and $v_k$ denote the 2D coordinate and the visibility of the $k$-th keypoint, respectively. We design two distinct token initialization ways as follows: i) for visible keypoints ($v_k = 1$), we use the Fourier embedding (Mildenhall et al., 2021) to map the 2D coordinate to the corresponding feature dimensions; ii) for invisible keypints ($v_k = 0$), we employ a shared learnable mask token (He et al., 2022b) to represent the invisible position. 2) ***Encoding Process of Keypoint Tokens.*** Since the initialized keypoint tokens only contain pixel-level position information, we further introduce two encoding mechanisms: i) the "keypoint token to keypoint token" attention to capture potential structural relations; ii) The "image patch token to keypoint token" attention to propagate global image feature information into each keypoint token.

## 2.2 CROSS-MODALITY INTERACTIVE ENCODER AND DECODER.

*UniPose* extends previous close-set keypoint detection to open-set scenarios through the incorporation of multi-modality prompts. To facilitate this, we introduce both the Cross-Modality Interactive Encoder and Decoder, allowing for interaction between the input image and multi-modality prompts, as shown in Fig. 5-(b) and (c).

**Cross-Modality Interactive Encoder.** In addition to the deformable self-attention layers for images employed in previous work (Shi et al., 2022; Yang et al., 2022a) *i.e.,* grey module of Fig. 5-(b), our Cross-Modality Interactive Encoder further introduces self-attention layers for prompts and interleaved cross-attention layers connecting images and prompts, as in blue modules of Fig. 5-(b).

**Cross-Modality Interactive Decoders.** *UniPose* decouples the decoder into two components: the instance-level decoder and the keypoint-level decoder. This separation allows for keypoint detection in a coarse-to-fine manner. In previous work, object queries and keypoint queries are used to independently query for corresponding bounding boxes and keypoints through self-attention between queries and image-to-query cross-attention, *i.e.,* grey module of Fig. 5-(c). To enhance prompt-guided keypoint detection, we take a step further by integrating prompt representations into the queries via prompt-to-query cross attention, as shown in blue modules of Fig. 5-(c).

## 2.3 TRAINING AND INFERENCE PIPELINE

We adopt the same bounding box and keypoint regression losses as previous end-to-end works (Yang et al., 2022a): the L1 loss and the GIOU loss (Rezatofighi et al., 2019) for object's bounding box regression $\mathcal{L}^{obj}_{reg}$; the L1 loss and the OKS loss (Shi et al., 2022) for keypoint regression $\mathcal{L}^{kpt}_{reg}$. In addition, *UniPose* replaces the object classification loss with Prompt-to-Object contrastive loss and introduces the Prompt-to-Keypoint contrastive loss for fine-grained alignment.

**Instance-level Alignment.** Previous keypoint detection frameworks mainly focus on close-set objects and typically use a simple linear layer as the object classifier. In contrast, *UniPose* encode multi-modality prompts (*i.e.*, text or image) into the corresponding object prompt tokens in a unified formulation $\widehat{\mathbf{F}}^t_{obj}, \widehat{\mathbf{F}}^i_{obj} \in \mathbb{R}^{L \times C}$, where $L$ is the number of object classes in prompts and $C$ indicates the feature dimension. Following (Li et al., 2022; Liu et al., 2023b), we employ contrastive loss between predicted objects $\mathbf{Q}^{\ddagger}_{obj}$ and prompt tokens for classification. More specifically, we

compute the dot product between each object query and the prompt tokens to predict logits for each token and then calculate the Focal loss of each logit $\mathcal{L}_{align}^{obj}$ for optimization.

**Keypoint-level Alignment.** In previous keypoint detection frameworks, the classification problem related to keypoints is often overlooked and the learning process mainly focuses on establishing a one-to-one mapping between predicted and labeled keypoints. In contrast, *UniPose* takes the first step toward Prompts-to-Keypoint alignment using a unified set of keypoint definitions. Similar to coarse-grained alignment, we can also obtain the keypoint prompt tokens in a unified formulation $\widehat{\mathbf{F}}_{kpt}^{t}, \widehat{\mathbf{F}}_{kpt}^{i} \in \mathbb{R}^{K \times C}$, where $K$ denotes the number of keypoint categories in prompts. We utilize contrastive loss between predicted keypoints $\mathbf{Q}_{kpt}^{\dagger}$ and prompt tokens for classification. To elaborate, we compute the dot product between each keypoint query and the prompt tokens to predict the logits for each token. Subsequently, we calculate the Focal loss for each logit $\mathcal{L}_{align}^{kpt}$ to optimize the model.

**The Overall Loss.** The overall training pipeline of *UniPose* can be written as follows,

$$\mathcal{L} = \mathcal{L}_{reg}^{obj} + \mathcal{L}_{reg}^{kpt} + \mathcal{L}_{align}^{obj} + \mathcal{L}_{align}^{kpt} \qquad (1)$$

**Inference Pipeline** 1) *Textual Prompts as inputs.* *UniPose* can utilize pre-defined object classes with keypoints definitions as text prompts to obtain quantitative results. In practical scenarios, users can provide prompts to predict the desired objects with keypoints. 2) *Visual Prompt as inputs.* *UniPose* can randomly sample a set of image prompts from the training data to obtain quantitative results. In practical scenarios, users can provide a single instance image with the corresponding keypoint definition to predict all the similar objects in the test images.

## 3 UNIKPT: A UNIFIED DATASET FOR KEYPOINT DETECTION

**Unifying 13 Keypoint Datasets into *UniKPT*.** Existing keypoint detection datasets have already concentrated on various object categories with specific pre-defined keypoints. However, several challenges still exist. 1) The majority of 2D keypoint detection datasets predominantly concentrate on human-related categories, such as human body, face, and hands. For other object categories, datasets are relatively scarce and fragmented. 2) Each dataset typically encompasses a single super-category of objects, each associated with one or a few sets of keypoint-defined skeletons.

Table 1: Statistics of *UniKPT* with 13 existing keypoint datasets. *KPT* and *Uni.* indicate keypoint and Unified, respectively.

| Datasets | KPT | Class | Images | Instances | Uni. Images | Uni. Instances |
|---|---|---|---|---|---|---|
| COCO | 17 | 1 | 58,945 | 156,165 | 58,945 | 156,165 |
| 300W-Face | 68 | 1 | 3,837 | 4,437 | 3,837 | 4,437 |
| OneHand10K | 21 | 1 | 11,703 | 11,289 | 2,000 | 2000 |
| Human-Art | 17 | 1 | 50,000 | 123,131 | 50,000 | 123,131 |
| AP-10K | 17 | 54 | 10,015 | 13,028 | 10,015 | 13,028 |
| APT-36K | 17 | 30 | 36,000 | 53,006 | 36,000 | 53,006 |
| MacaquePose | 17 | 1 | 13,083 | 16,393 | 2,000 | 2,320 |
| Animal Kingdom | 23 | 850 | 33,099 | 33,099 | 33,099 | 33,099 |
| AnimalWeb | 9 | 332 | 22,451 | 21,921 | 22,451 | 21,921 |
| Vinegar Fly | 31 | 1 | 1,500 | 1,500 | 1,500 | 1,500 |
| Desert Locust | 34 | 1 | 700 | 700 | 700 | 700 |
| Keypoint-5 | 55/31[1] | 5 | 8,649 | 8,649 | 2,000 | 2,000 |
| MP-100 | 561/293[1] | 100 | 16,943 | 18,000 | 16,943 | 18,000 |
| UniKPT | 338 | 1237 | - | - | 226,547 | 418,487 |

[1] Keypoint-5 and MP-100 have different categories with varying numbers of keypoints. While the cumulative count of keypoints reaches 55 and 561 by aggregating across categories, we consolidate them into unified counts of 31 and 293 keypoints by leveraging textual descriptions.

As a result, there is currently no generalist model capable of achieving keypoint detection across all possible scenarios. Motivated by these, we propose to unify existing keypoint detection datasets based on three principles: i) collecting and encompassing all articulated, rigid, and soft objects, ii) including a broader spectrum of object categories whenever possible, and iii) spanning a diverse range of image styles. As shown in Table. 1, we have unified 13 keypoint detection datasets, including COCO (Lin et al., 2014), 300W-Face (Sagonas et al., 2016), OneHand10K (Wang et al., 2018), Human-Art (Ju et al., 2023), AP-10K (Yu et al., 2021), APT-36K (Yang et al., 2022b), MacaquePose (Labuguen et al., 2021), Animal Kingdom (Ng et al., 2022), AnimalWeb (Khan et al., 2020), Vinegar Fly (Pereira et al., 2019), Desert Locust (Graving et al., 2019), Keypoint-5 (Wu et al., 2016), and MP-100 (Xu et al., 2022a). It is worth noting that MP-100 also includes training subsets from two other datasets, Deepfashion2 (Ge et al., 2019) and Carfusion (Reddy et al., 2018).

**Statistical Analysis.** In total, the unified dataset comprises $226,547$ images and $418,487$ instances, featuring 338 keypoints and $1,237$ categories. In particular, for articulated objects like humans and animals, we further categorize them based on biological taxonomy, resulting in $1,216$ species, 66 families, 23 orders, and 7 classes.

## 4 EXPERIMENT

Due to the page limit, we leave the detailed experiment setup, data organization, and more experiments in the Appendix.

### 4.1 UNSEEN OBJECTS AND KEYPOINTS DETECTION

We evaluate *UniPose* against the previous methods, *i.e.*, ProtoNet (Snell et al., 2017), MAML (Finn et al., 2017), Fine-tune (Nakamura & Harada, 2019), POMNet (Xu et al., 2022a), and Cape-former (Shi et al., 2023) on the MP-100 dataset in Tab. 2, to demonstrate is generalization abilities for both unseen object and keypoint detection. **First**, with ground-truth bounding boxes (excluding the challenge of generalization to unseen objects), *UniPose*, as an end-to-end framework, achieves state-of-the-art results, surpassing all top-down methods, and offers efficiency by requiring only a single forward pass for scenes with multiple objects. **Second**, in the absence of ground-truth bounding boxes, *UniPose* exhibits a significant improvement over CapeFormer in terms of average PCK, achieving a significant increase of 42.8%, thanks to *UniPose*'s generalization ability for both unseen object and keypoint detection. Furthermore, we distinguish between single-object and multi-object scenes in the test set, as shown in Tab. 3 and Tab. 12. *UniPose*'s advantages are particularly pronounced in multi-object scenes. Notably, CapeFormer exhibits sensitivity to input resolution, with a sharp performance drop when increasing resolution from 256 to 800.

Table 2: Comparisons of visual prompt-based keypoint detection for unseen objects and keypoints using the MP-100 dataset. **TD** and **E2E** refer to the top-down and end-to-end paradigms, respectively. The inference times for all methods are tested on an A100 with a batch size of 1. Top-down methods need multiple inferences when $N$ objects are detected in an image.

|  | Method | Backbone | Input Image | Box Anno | Split1 | Split2 | Split3 | Split4 | Split5 | Mean (PCK) | Time [ms] |
|---|---|---|---|---|---|---|---|---|---|---|---|
| **TD** | ProtoNet | ResNet-50 | Cropped | ✓ | 46.05 | 40.84 | 49.13 | 43.34 | 44.54 | 44.78 | - |
|  | MAML | ResNet-50 | Cropped | ✓ | 68.14 | 54.72 | 64.19 | 63.24 | 57.20 | 61.50 | - |
|  | Fine-tune | ResNet-50 | Cropped | ✓ | 70.60 | 57.04 | 66.06 | 65.00 | 59.20 | 63.58 | - |
|  | POMNet | ResNet-50 | Cropped | ✓ | 84.23 | 78.25 | 78.17 | 78.68 | 79.17 | 79.70 | $151 \times N$ |
|  | CapeFormer | ResNet-50 | Cropped | ✓ | **89.45** | 84.88 | 83.59 | 83.53 | 85.09 | 85.31 | $57 \times N$ |
|  | CapeFormer | ResNet-50 | Original | ✗ | 60.74 | 57.18 | 54.04 | 46.53 | 42.35 | 52.17 | $57 \times N$ |
| **E2E** | *UniPose* | ResNet-50 | Original | ✓ | 89.07 | **85.05** | **85.26** | **85.52** | **85.79** | **86.14** | **59** |
|  | *UniPose* | ResNet-50 | Original | ✗ | 76.47 | 72.16 | 71.57 | 75.89 | 76.43 | 74.50 | **59** |

Note: We train our models only on the MP-100 dataset to ensure a fair comparison. During evaluation, all methods use the same visual prompts paired with test images.

Table 3: Comparisons on the specific *multi-object* MP-100 test set.

| Methods | Backbone | Input Image | Resolution | Split1 | Split2 | Split3 | Split4 | Split5 | Mean (PCK) |
|---|---|---|---|---|---|---|---|---|---|
| CapeFormer | ResNet-50 | Original | 256×256 | 24.19 | 23.81 | 24.39 | 21.21 | 20.30 | 22.78 |
| CapeFormer | ResNet-50 | Original | 800×800 | 24.53 | 30.52 | 17.19 | 20.90 | 20.59 | 28.75 |
| *UniPose* | ResNet-50 | Original | 800×800 | **69.40** | **66.49** | **64.44** | **63.95** | **63.28** | **65.51** |

### 4.2 COMPARISON WITH SOTA EXPERT KEYPOINT DETECTION MODELS

**Generic Keypoint Detection.** We present a comparative analysis of *UniPose* against state-of-the-art models that have been trained on multiple datasets, ViTPose++ (Xu et al., 2022c) and ED-pose (Yang et al., 2022a). Our evaluation benchmarks 12 datasets as shown in Tab. 5. The results demonstrate that *UniPose* consistently delivers superior performance across all datasets. Notably, when compared to ViTPose++, which lacks the capability to handle unseen datasets with different keypoint structures, *UniPose* excels by detecting more objects and keypoints in an end-to-end manner.

**Comparison with Baseline (ED-Pose) Aligned with Training Data.** *UniPose* is built on ED-Pose in a coarse-to-fine keypoint detection approach. Here,

Table 4: Comparisons with baseline-ED-Pose under a fair multi-dataset training setting, using the Swin-T backbone.

| Methods | Instance-level | | Keypoint-level | | |
|---|---|---|---|---|---|
|  | $AP_M$ | $AP_L$ | AP | $AP_M$ | $AP_L$ |
| *COCO* `val set` | | | | | |
| ED-Pose | 68.8 | 79.0 | 73.3 | 67.6 | 81.5 |
| *UniPose-T* | **71.1** | 80.2 | **74.2** | **68.8** | **82.1** |
| *UniPose-V* | **71.1** | **80.3** | 74.1 | **68.8** | 81.8 |
| *Human-Art* `val set` | | | | | |
| ED-Pose | 32.3 | 61.5 | 71.3 | 37.2 | 75.9 |
| *UniPose-T* | 33.7 | **63.1** | **72.2** | **39.5** | **76.7** |
| *UniPose-V* | **34.0** | 63.0 | 71.8 | 39.3 | 76.4 |
| *AP-10K* `val set` | | | | | |
| ED-Pose | 53.7 | 62.5 | 45.5 | 31.0 | 46.5 |
| *UniPose-T* | 54.5 | 78.8 | **73.2** | 45.6 | **74.3** |
| *UniPose-V* | **55.8** | **79.0** | 72.8 | **47.2** | 74.0 |

we train both our *UniPose* and ED-Pose using the same datasets, *i.e.*, COCO, Human-Art, AP-10K, and APT-36K. The results in Tab. 4 show that *UniPose* outperforms ED-Pose across all datasets in terms of both instance-level and keypoint-level detection. Moreover, for the AP-10K dataset,

Table 5: Comparison with SOTA expert models trained on multiple datasets. † indicates results using the flipping test. Results marked with * rely on ground-truth bounding boxes for top-down methods. The expert models can test datasets with known keypoint structures, highlighted in blue, but cannot handle unseen datasets with different keypoint structures. We highlight the trained datasets in dark blue of expert models in *UniKPT*. The **best** results are highlighted in **bold**, and the second best results are highlighted with a underline. *T* and *V* denote textual and visual prompts used.

| Methods | Backbone | COCO | AP-10K | Human-Art AP↑ | Macaque | 300W | Hand | AK | Fly | Locust PCK@0.2↑ | KPT-5 | DF2 | Carfusion |
|---------|----------|------|--------|---------------|---------|------|------|-----|-----|--------|-------|-----|-----------|
| *Expert Models* | | | | | | | | | | | | | |
| ViTPose++† (TP) | ViT-S (MAE) | 75.8 | 71.4* | 23.4 | 15.5* | 95.2* | 96.1* | - | - | - | - | - | - |
| ViTPose++† (TP) | ViT-L (MAE) | **78.6** | **80.4*** | 35.6 | 51.9* | **99.8*** | 99.5* | - | - | - | - | - | - |
| ED-Pose (E2E) | Swin-T | 73.3 | 45.5 | 71.3 | 51.0 | - | - | - | - | - | - | - | - |
| *Prompted-based Models* | | | | | | | | | | | | | |
| *UniPose-T* (E2E) | Swin-T | 74.4 | 74.0 | 72.5 | 78.0 | 98.1 | 95.7 | 67.8 | 99.6 | 99.7 | 94.3 | 95.7 | 78.1 |
| *UniPose-V* (E2E) | Swin-T | 74.3 | 73.6 | 72.1 | 77.3 | 99.4 | 95.9 | 66.2 | 99.8 | 99.6 | 87.4 | 91.0 | 72.1 |
| *UniPose-T* (E2E) | Swin-L | 76.8 | 79.2 | **75.9** | 79.4 | 98.5 | **99.8** | **71.7** | **99.9** | 99.8 | **95.5** | **97.5** | **88.7** |
| *UniPose-V* (E2E) | Swin-L | 76.6 | 79.0 | 75.5 | 77.8 | 99.3 | 99.5 | 70.4 | **99.9** | **99.9** | 91.6 | 95.5 | 85.0 |

[1] Due to the absence of official train/val/test splits in AnimalWeb and APT-36K, we solely utilize them for training and do not conduct comparisons with other methods.
[2] ViTPose++: COCO + COCO-W + MPII + AIC + AP-10K + APT-36K, 387K training data.
[3] ED-Pose: COCO + Human-Art + AP-10K + APT-36K, 154K training data
[4] *UniPose*: UniKPT, 227K training data

which involves the classification of 54 different species, *UniPose* surpasses ED-Pose with a 27.7 AP improvement, thanks to instance-level and keypoint-level alignments.

**Qualitative Results on Existing Datasets.** Given an input image and textual prompts, *UniPose* can perform well for any articulated, rigid, and soft objects, as shown in Fig. 6.

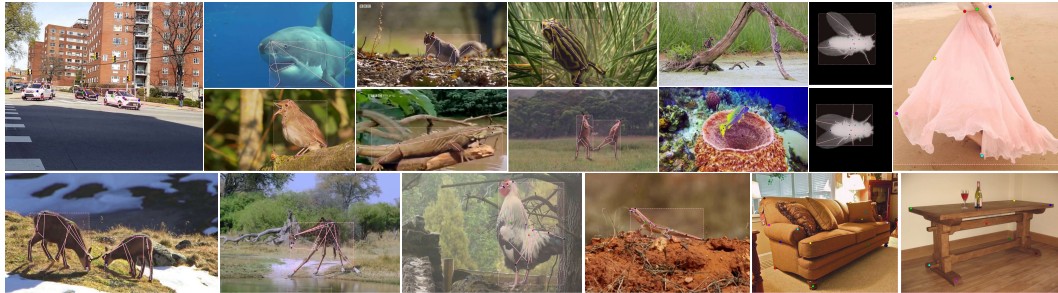

Figure 6: Visualization of the detected keypoints via *UniPose* on the unified dataset (*UniKPT*).

### 4.3 COMPARISON WITH GENERALIST MODELS FOR GENERIC KEYPOINT DETECTION

We compare our *UniPose* with generalist models Unified-IO (Lu et al., 2022), Painter (Wang et al., 2023), and InstructDiffsuion (Geng et al., 2023b) in terms of keypoint detection task. As shown in Tab. 6, *UniPose* outperforms all the generalist models across all evaluated datasets, which demonstrates *UniPose*'s capability to serve as a robust generalist keypoint detector.

Table 6: Comparisons with generalist models.

| Method | COCO `val` | HumanArt `val` | AP-10K `val` |
|--------|------------|----------------|--------------|
| Unified-IO | 25.0 | 15.7 | 7.6 |
| Painter | 70.2 | 12.4 | 15.3 |
| InstructDiffusion | 71.2 | 51.4 | 15.9 |
| *UniPose* (Image) | 76.6 | 75.5 | 79.0 |
| *UniPose* (Text) | **76.8** | **75.9** | **79.2** |

Table 7: Comparison of CLIP score.

| Methods | AP-10K `val` | | Human-Art `val` | |
|---------|-------------|----------|-----------------|----------|
| | Instance | Keypoint | Instance | Keypoint |
| CLIP | 28.36 | 21.75 | 23.60 | 23.81 |
| *UniPose* | **58.59** | **66.01↑** | **68.41** | **63.46** |

### 4.4 COMPARED WITH OPEN-VOCABULARY MODELS

**Comparison with Vision-Language Model-CLIP.** We assess *UniPose*'s text-to-image alignment capabilities at different granularities using instance descriptions and keypoint descriptions. As in Fig. 7, we report the CLIP score of *UniPose* and CLIP on AP-10K, which involves 54 animal categories, and Human-Art, which features 15 image styles. The results show that *UniPose* consistently provides higher-quality text-to-image similarity scores, both at the instance level and keypoint level.

**Comparison with Open-Vocabulary Detection Model.** We compare *UniPose* with the state-of-the-art open-vocabulary object detector, Grounding-DINO, in terms of instance-level and keypoint-level detection. We present the COCO results in Tab. 8, while results for other datasets are in Tab. 16. *UniPose* achieves comparable instance detection performance to the fine-tuned Grounding-DINO model. More importantly, Grounding-DINO fails to localize fine-grained keypoints, *UniPose* successfully addresses these challenges, achieving significant performance across all datasets.

Table 8: Comparisons with the state-of-the-art open-vocabulary object detector, focusing on instance-level and keypoint-level detection. ‡ denotes the fine-tuning of GroundingDINO using the keypoint detection datasets. Note that we limit the instance-level comparison to $AP_M$ (medium objects) and $AP_L$ (large objects), as small objects do not have keypoints annotated.

| Methods | Backbone | Instance-level | | Keypoint-level | | | Training Datasets | Dataset Volume |
|---|---|---|---|---|---|---|---|---|
| | | $AP_M$ | $AP_L$ | AP | $AP_M$ | $AP_L$ | | |
| *COCO val set* | | | | | | | | |
| GroundingDINO-T | Swin-T | 70.8 | 82.0 | 3.1 | 2.8 | 3.2 | O365,GoldG,Cap4M | 1858K |
| GroundingDINO-T | Swin-B | 69.7 | 79.5 | 6.8 | 6.6 | 7.2 | COCO,O365,GoldG,Cap4M,OpenImage,ODinW-35,RefCOCO | - |
| GroundingDINO‡-T | Swin-T | **71.2** | **83.4** | 1.8 | 1.7 | 1.9 | COCO,Human-Art,AP-10K,APT-36K | 1858K + 155K |
| *UniPose*-T | Swin-T | 71.1 | 80.2 | **74.2** | **68.8** | **82.1** | COCO,Human-Art,AP-10K,APT-36K | 155K |
| *UniPose*-V | Swin-T | 71.1 | 80.3 | 74.1 | **68.8** | 81.8 | COCO,Human-Art,AP-10K,APT-36K | 155K |

## 4.5 ABLATION STUDY

In this section, we **firstly** validate the effectiveness of the *UniPose* framework in instance-to-keypoint alignment and multi-modality prompts. We train *UniPose* with the Swin-T backbone on four datasets: COCO, Human-Art, AP-10K, and APT36K. For comparison, we report the results on AP-10K, which encompasses multiple object categories and enables a comprehensive evaluation in classification and localization. **Secondly**, we assess the effectiveness of the *UniKPT*'s data by scaling up the dataset. Similarly, the Swin-T backbone is adopted. We present the results on both the seen dataset AP-10K in *UniKPT* and the unseen dataset AnimalPose (Cao et al., 2019) to demonstrate its generalization ability.

**Instance-to-Keypoint Alignment.** As discussed in Sec. 2.3, we introduce $\mathcal{L}_{Align}^{obj}$ and $\mathcal{L}_{Align}^{kpt}$ to facilitate prompt-to-instance and prompt-to-keypoint alignment, respectively. We present the results via textual prompts in Tab. 9, highlighting the significant improvement in detection performance, particularly in $AP_L$, due to $\mathcal{L}_{Align}^{obj}$. This underscores its importance in aiding the model to distinguish between categories and enhance classification performance. The improved detection performance positively affects keypoint performance. Moreover, the inclusion of $\mathcal{L}_{Align}^{kpt}$ further helps the network learn keypoint distinctions, resulting in enhanced keypoint detection performance."

Table 9: Impact of instance-to-keypoint alignment on AP-10K.

| $\mathcal{L}_{Align}^{obj}$ | $\mathcal{L}_{Align}^{kpt}$ | Instance-level | | Keypoint-level | | |
|---|---|---|---|---|---|---|
| | | $AP_M$ | $AP_L$ | AP | $AP_M$ | $AP_L$ |
| | | 53.7 | 62.5 | 45.5 | 31.0 | 46.5 |
| ✓ | | 53.8 | 78.5 | 72.6 | 43.6 | 73.4 |
| ✓ | ✓ | **54.5** | **78.8** | **73.2** | **45.6** | **74.3** |

Table 10: Impact of two modal prompts on AP-10K. The prompt used in the test is highlighted in grey.

| Visual Prompt | Textual Prompt | Instance-level | | Keypoint-level | | |
|---|---|---|---|---|---|---|
| | | $AP_M$ | $AP_L$ | AP | $AP_M$ | $AP_L$ |
| ✓ | | 53.3 | 78.1 | 71.5 | 43.4 | 72.4 |
| ✓ | ✓ | **55.8** | **79.0** | 72.8 | **47.2** | 74.0 |
| | ✓ | 53.8 | 78.5 | 72.9 | 45.1 | 74.2 |
| ✓ | ✓ | 54.5 | 78.8 | **73.2** | 45.6 | **74.3** |

**Multi-Modlity Prompts.** We utilize both the visual and textual prompts by default during training. Here, we perform an ablation study by removing one of these prompts, as depicted in Tab. 10. The results highlight the mutual advantages of both textual and visual prompts.

**Impact on Dataset Quantity.** We first train our *UniPose* using 4 datasets covering humans and 60 different animals. Then, we add additional 5 animal datasets to train *UniPose*, as shown in Tab. 11. This results in significant improvements in both instance and keypoint detection on seen AP-10K datasets (using textual prompts). Moreover, we achieve a significant improvement on the unseen AnimalPose dataset (using visual prompts), thanks to the broader range of categories and the increased data size. Furthermore, we incorporate additional part-level datasets (Face and Hand) as well as rigid and soft object datasets for training. Although these diverse datasets lead to a slight decrease in AP-10K performance, it further boosts the model's performance on unseen datasets.

Table 11: Impact of dataset quantity on AP-10K and AnimalPose.

| Training Data | AP-10K's Instance | | AP-10K's Keypoint | | | AnimalPose |
|---|---|---|---|---|---|---|
| | $AP_M$ | $AP_L$ | AP | $AP_M$ | $AP_L$ | PCK |
| COCO,Human-Art,AP-10K,APT-36K | 54.5 | 78.8 | 73.2 | 45.6 | 74.3 | 52.7 |
| +MacquePose,AnimalKingdom,AnimalWeb,Vinegar Fly,Desert Locust | **55.6** | **80.2** | **74.2** | **48.3** | **75.0** | 70.1 |
| +300w-Face,OneHand10K,Keypoint-5,MP-100 | 55.3 | 78.8 | 74.0 | 47.8 | 74.7 | **73.4** |

## 5 CONCLUSION

This work studies the problem of detecting any keypoints from instance to keypoint levels via either visual prompts or textual prompts. To solve this problem, we proposed an end-to-end coarse-to-fine framework trained on a unified keypoint dataset to learn general semantic fine-grained keypoint concepts and global-to-local keypoint structure, achieving high performance and generalizability. We leave broader impact and limitation discussions in the Appendix D.

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

# Appendix:
# UniPose: Detecting Any Keypoints

## A    RELATED WORK

### A.1    CATEGORY-SPECIFIC KEYPOINT DETECTION

Keypoint detection is a fundamental computer vision task for fine-grained image understanding, aiming to localize the pre-defined semantic keypoints of objects within an image. For a long time, mainstream methods have focused on multi-person or animal pose estimation (Sun et al., 2023a; Ye et al., 2022; Mathis et al., 2018; Ng et al., 2022; Xu et al., 2022b;c; Jiang et al., 2023; Yang et al., 2022a). Approaches of multi-person pose estimation can be categorized into two-stage and one-stage methods. Among the two-stage methods, there are top-down and bottom-up strategies. Top-down methods (Xiao et al., 2018; Sun et al., 2019; Li et al., 2021; Mao et al., 2022; Xu et al., 2022b; Geng et al., 2023a) demonstrate impressive performance but come with a higher computational cost. They first detect each person in an image using an independent object detector and then perform single-person pose estimation with the proposed model. In contrast, bottom-up methods (Cao et al., 2017; Newell et al., 2017; Cheng et al., 2020; Luo et al., 2021; Geng et al., 2021) are more efficient but have lower precision. They begin by estimating keypoints and then group them into individual human poses. Specifically, one-stage end-to-end methods (Shi et al., 2022; Yang et al., 2022a; Liu et al., 2023a; Yang et al., 2023a) have shown superior performance and efficiency trade-offs. ED-Pose (Yang et al., 2022a) introduces a coarse-to-fine framework that explicitly incorporates human detection into the fine-grained keypoint detection process. However, mainstream methods primarily focus on single super-category objects with specific pre-defined keypoints.

To detect more keypoints, training multiple models is straightforward to handle various categories with different keypoint definitions (Xu et al., 2022c; Contributors, 2020; Zauss et al., 2021). Based on the given instance-level detection and multiple datasets training, the two-stage top-down method ViTPose+ achieves state-of-the-art human and animal keypoint detection performance. However, the top-down strategy still requires reliance on corresponding object detectors and faces challenges in handling missing object detections and the high computational costs for crowded scenes. Additionally, existing ViTPose++ could only support the keypoint detection from the trained 61 (*e.g.*, person and 60 animal species) categories. Furthermore, although there are some vision generalist models (Chen et al., 2022; Lu et al., 2022; Wang et al., 2023; Geng et al., 2023b) that have been employed to address various vision tasks, including keypoint detection, all of them are primarily designed for human keypoint detection and will fail on other type keypoint of objects or unseen keypoints. In light of these, our work aims to provide a more powerful end-to-end keypoint generalist via unifying existing keypoint detection tasks by leveraging 13 keypoint detection datasets.

### A.2    CATEGORY-AGNOSTIC KEYPOINT DETECTION

Given a prompt image of a novel object and its corresponding keypoint definition, visual prompt-based keypoint detection/pose estimation aims to detect category-agnostic keypoints. Existing works (Xu et al., 2022a; Shi et al., 2023) simplify this problem as a single-object keypoint matching problem and train their models on a small-scale dataset (MP-100 with 17K images spanning 100 categories), making the model suffer from under-fitting and hard to learn the local keypoint representation effectively. The concurrent work UniAP (Sun et al., 2023a) unifies animal pose estimation, segmentation, and classification under a single model via few-shot learning. It still follows the single-object detection and only supports visual prompts as input. In contrast, we introduce an end-to-end prompt-based keypoint detection framework that can detect multi-object keypoints and unify the existing 13 datasets to generalize the model across instances and their keypoints.

### A.3    OPEN-VOCABULARY VISION MODELS

Benefit from vision-language pretrained model CLIP (Radford et al., 2021), like open-vocabulary object detection and semantic segmentation tasks are actively explored (Zang et al., 2022; Gu et al., 2021; Li et al., 2022; Yao et al., 2022; Liu et al., 2023b; Liang et al., 2023; Zhong et al., 2022), especially for a finer level of granularity understanding in image-text pairs (Li et al., 2023; Sun

et al., 2023b). The most related work is CLAMP (Zhang et al., 2023). It leverages CLIP with language guidance to prompt the animal keypoints. However, *open-vocabulary* keypoint detection is still under-explored, which can distinguish orientation information and local structure relations in a compact and fine-grained representation.

# B    UNIKPT DATASET

According to Fig. 7, we observe that each dataset only focuses on a single super-category (e.g., "human only" and "animal only"), making it challenging to achieve keypoint generalization when using them individually. Additionally, there are significant differences in the quality, quantity, and appearance styles of keypoint annotations in these datasets. Therefore, we are motivated to unify all the datasets to train a generalist keypoint detector.

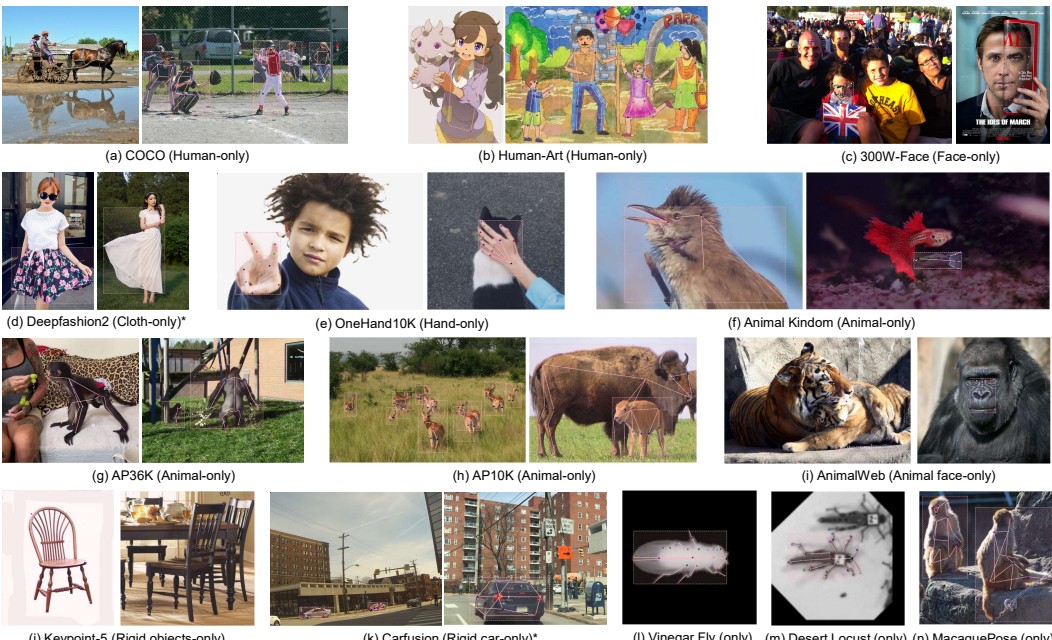

Figure 7: Visualization the unified dataset (*UniKPT*) for each original dataset. * means the two datasets are included in MP-100 (Xu et al., 2022a).

## B.1    GUIDELINES FOR UNIFYING DATASETS

**Unifying Keypoint's Textual Description.** Given that descriptions for the same keypoint may vary across different datasets, our first step is to standardize the names of overlapping keypoints. Then we could align fine-grained keypoint features that share similar local visual patterns, such as *"the left eye of humans"* and *"the left eye of cats"*, by using a unified textual description.

**Unifying Orientation Criterion.** We observe discrepancies in the existing data regarding the definition of left and right orientations. Therefore, we standardize left and right orientation definitions based on the object's left and right within the image, which can promote the alignment of directional information between the text and keypoints within the model.

**Annotating Detailed Keypoint Information.** The majority of keypoints in the unified dataset lack detailed annotations to distinguish between different keypoints, such as the keypoints in facial contours or cloth landmarks. To benefit the fine-grained text-to-keypoint alignment, we further provided detailed names for all keypoints, which include rich directional information (e.g., upper, lower, left, right) and descriptive terminology.

**Balancing Data Volume and Object Categories.** Among the 13 datasets, we have used the entire dataset for most of them because they encompass a diverse range of multi-object scenarios (e.g.,

Table 12: Comparisons with the state-of-the-art method on the MP-100 dataset, differentiated between single-object scene and multi-object scene.

| Methods | Backbone | Input Image | Resolution | Split1 | Split2 | Split3 | Split4 | Split5 | Mean (PCK) |
|---|---|---|---|---|---|---|---|---|---|
| *All the images in the* `test set` | | | | | | | | | |
| CapeFormer | ResNet-50 | Original | 256×256 | 60.74 | 57.18 | 54.04 | 46.53 | 42.35 | 52.17 |
| CapeFormer | ResNet-50 | Original | 800×800 | 44.32 | 36.68 | 32.63 | 39.35 | 36.18 | 37.83 |
| *UniPose* | ResNet-50 | Original | 800×800 | **76.47** | **72.16** | **71.57** | **75.89** | **76.43** | **74.50** |
| *Single-object images in the* `test set` | | | | | | | | | |
| CapeFormer | ResNet-50 | Original | 256×256 | 63.27 | 58.40 | 56.16 | 48.53 | 43.26 | 53.92 |
| CapeFormer | ResNet-50 | Original | 800×800 | 45.69 | 36.90 | 33.73 | 40.80 | 36.83 | 38.79 |
| *UniPose* | ResNet-50 | Original | 800×800 | **77.66** | **72.73** | **72.08** | **76.63** | **77.47** | **75.31** |
| *Multi-object images in the* `test set` | | | | | | | | | |
| CapeFormer | ResNet-50 | Original | 256×256 | 24.19 | 23.81 | 24.39 | 21.21 | 20.30 | 22.78 |
| CapeFormer | ResNet-50 | Original | 800×800 | 24.53 | 30.52 | 17.19 | 20.90 | 20.59 | 28.75 |
| *UniPose* | ResNet-50 | Original | 800×800 | **69.40** | **66.49** | **64.44** | **63.95** | **63.28** | **65.51** |

COCO), varied photo styles (e.g., Human-Art), or a large number of species categories (e.g., AnimalKindom). However, for some datasets such as Onehand10K for human hand, MacaquePose for macaque animals, and Keypoint-5 for furniture, each only containing single-object scenes featuring limited categories, we randomly sample 2,000 training images from each for model training.

# C EXPERIMENTS

## C.1 EXPERIMENTAL SETUP

**Dataset.** We conduct our experiments in two settings: 1) **MP-100 Dataset** (Xu et al., 2022a): We employ the MP-100 dataset to conduct a fair evaluation of *UniPose*'s generalization capabilities against previous visual prompt-based approaches. This dataset comprises approximately 17K images and 18K instances spanning 100 different categories. The number of keypoints varies across categories, ranging from 8 to 68. To facilitate model training and evaluation, the dataset is divided into five splits. Each split ensures that the training, validation, and test categories are non-overlapping, *i.e.*, the categories for evaluation are not accessed during training. 2) **Unified Dataset - *UniKPT*** (See Table. 1): we employ *UniKPT* to train *UniPose* and subsequently evaluate it on 12 datasets' evaluation sets, *i.e.*,. COCO, Human-Art, AP-10K, MacaquePose (Macaque), 300W, One-Hand10K (Hand), Animal Kingdom (AK), Vinegar Fly (Fly), Desert Locust (Locust), Keypoint-5 (KPT-5), Deepfashion2 (D2), and Carfusion. Notably, it's essential to mention that due to the lack of official train/val/test splits in AnimalWeb and APT-36K, we exclusively employ them for training purposes and refrain from conducting comparisons with other methods.

**Evaluation Metric.** 1) **PCK Metric for MP-100 Dataset**: we employ the Probability of Correct Keypoint (PCK) (Yang & Ramanan, 2012) as the quantitative metric. The threshold for PCK is set to 0.2 following POMNet (Xu et al., 2022a) and Capeformer (Shi et al., 2023). 2) **AP and PCK Metrics for the unified dataset *UniKPT***: In accordance with the standard settings specific to each dataset, we employ two evaluation metrics, *i.e.*, OKS-based Average Precision (AP) (Lin et al., 2014) and PCK with a threshold of 0.2.

**Implementation details.** We use the exact same training details as all the end-to-end models (Yang et al., 2022a; Shi et al., 2022). Specifically, we augment the training images through random cropping, flipping, and resizing. The shorter sides are kept within $[480, 800]$, while the longer sides are less than or equal to 1333. We utilize the AdamW optimizer with a weight decay of 1e-4. Our models are trained on 8 Nvidia A100 GPUs with a batch size of 16. During testing, the images are resized with shorter sides of 800 and longer sides less than or equal to 1333.

## C.2 MORE RESULTS

**Compared with Expert Models.** We present the complete results on three mainstream datasets, *i.e.*. COCO in Fig. 13, Human-Art in Fig. 14 and AP-10K in Fig. 15. The results demonstrate that *UniPose* establishes a new state-of-the-art benchmark among end-to-end models. Remarkably, *UniPose* also achieves results that are comparable to those of SOTA top-down methods.

**Compared with Open-Vocabulary Detection Model.** We compare *UniPose* with the state-of-the-art open-vocabulary object detector, Grounding-DINO, with a specific focus on instance-level and

Table 13: Comparisons with state-of-the-art methods on COCO `val2017` dataset. † denotes the flipping test.

| | Method | Backbone | AP | $AP_{50}$ | $AP_{75}$ | $AP_M$ | $AP_L$ |
|---|---|---|---|---|---|---|---|
| **TD** | HRNet† | HRNet-w32 | 74.4 | 90.5 | 81.9 | 70.8 | 81.0 |
| | HRNet† | HRNet-w48 | 75.1 | 90.6 | 82.2 | 71.5 | 81.8 |
| | ViTPose-S† | ViT-S (MAE) | 73.8 | 90.3 | 81.3 | 67.1 | 75.8 |
| | ViTPose-L† | ViT-L (MAE) | 78.3 | 91.4 | 85.2 | 71.0 | 81.1 |
| | ViTPose++-L† | ViT-L (MAE) | **78.6** | 91.4 | **85.4** | 71.5 | 81.3 |
| **E2E** | PETR | Swin-L | 73.1 | 90.7 | 80.9 | 67.2 | 81.7 |
| | ED-Pose | Swin-T | 73.6 | 90.9 | 80.6 | 68.3 | 81.3 |
| | ED-Pose | Swin-L | 74.3 | 91.5 | 81.6 | 68.6 | 82.6 |
| | *UniPose* (Text) | Swin-T | 74.4 | 90.7 | 81.0 | 68.8 | 82.1 |
| | *UniPose* (Image) | Swin-T | 74.3 | 90.6 | 81.0 | 68.8 | 81.8 |
| | *UniPose* (Text) | Swin-L | 76.8 | **91.9** | 83.8 | **71.6** | **84.8** |
| | *UniPose* (Image) | Swin-L | 76.6 | 91.8 | 83.6 | 71.5 | 84.5 |

HRNet, ViTPose, PETR, ED-Pose: COCO, 58K training data
ViTPose++: COCO+ COCO-W + MPII + AIC + AP-10K + APT-36K, 387K training data
*UniPose*: *UniKPT*, 226K training data

Table 14: Comparisons with state-of-the-art methods on Human-Art `val` dataset. † denotes the flipping test.

| | Method | Backbone | AP | $AP_{50}$ | $AP_{75}$ | $AP_M$ | $AP_L$ |
|---|---|---|---|---|---|---|---|
| **TD** | HRNet† | HRNet-w32 | 39.9 | 54.5 | 42.0 | 46.6 | 61.3 |
| | HRNet† | HRNet-w48 | 41.7 | 55.3 | 44.2 | 48.1 | 61.7 |
| | ViTPose-S† | ViT-S (MAE) | 38.1 | 53.2 | 40.5 | 44.8 | 60.2 |
| | ViTPose-L† | ViT-L (MAE) | 45.9 | 59.2 | 48.7 | 52.5 | 65.6 |
| **E2E** | ED-Pose | Swin-T | 71.3 | 85.6 | 77.0 | 37.2 | 75.9 |
| | *UniPose* (Text) | Swin-T | 72.5 | 86.7 | 77.6 | 39.5 | 76.7 |
| | *UniPose* (Image) | Swin-T | 72.1 | 86.5 | 77.0 | 39.3 | 76.4 |
| | *UniPose* (Text) | Swin-L | **75.9** | **89.6** | **81.7** | **42.6** | **80.1** |
| | *UniPose* (Image) | Swin-L | 75.5 | 89.5 | 81.5 | 42.2 | 79.7 |

HRNet, ViTPose: Human-Art, 58K training data
ED-Pose: COCO + Human-Art + AP-10K + APT-36K, 154K training data
*UniPose*: *UniKPT*, 226K training data

keypoint-level detection in Fig. 16. **For instance detection,** the original Grounding-DINO performs admirably on COCO since it has knowledge of the person categories. However, its performance sharply drops when the image style shifts to artificial scenes and when detecting the 54 different animal categories. After fine-tuning Grounding-DINO using the keypoint detection datasets, its detection performance on Human-Art and AP-10K has significantly improved. *UniPose* has also achieved comparable detection performance to the fine-tuned Grounding-DINO model. **For keypoint detection**, while Grounding-DINO fails to localize fine-grained keypoint, *UniPose* successfully addresses these challenges, achieving significant performance across all datasets.

## C.3 FAILURE CASE ANALYSIS

In Fig. 8, we show three kinds of failure cases in our method. First, due to multi-object and multi-keypoint contrastive learning with textual prompts, the classification scores of instances and keypoints may be imbalance across different categories. It will cause that it is hard to set a certain threshold for precise detection. From the left figure of Fig. 8(a), we set a higher threshold to demonstrate the outputs, and some objects are missing; we set a lower threshold, which will lead to some redundant detection. Second, despite the fact that we collected as many categories as possible, there are some keypoints of soft objects (*e.g.*, marine organisms shown in Fig. 8(b)) that are still difficult to

Table 15: Comparisons with state-of-the-art methods on AP-10K `val` dataset. † denotes the flipping test. **All the top-down methods are based on ground-truth boxes.**

| | Method | Backbone | AP | $AP_{50}$ | $AP_{75}$ | $AP_M$ | $AP_L$ |
|---|---|---|---|---|---|---|---|
| **TD** | HRNet† | HRNet-w32 | 72.2 | 93.9 | 78.7 | 55.5 | 73.0 |
| | HRNet† | HRNet-w48 | 73.1 | 93.7 | 80.4 | 57.4 | 73.8 |
| | ViTPose-S++† | ViT-S (MAE) | 71.4 | 93.3 | 78.4 | 47.6 | 71.8 |
| | ViTPose-L++† | ViT-L (MAE) | 80.4 | 97.6 | 88.5 | 52.7 | 80.8 |
| | ED-Pose | Swin-T | 45.5 | 57.4 | 50.2 | 31.0 | 46.5 |
| **E2E** | *UniPose* (Text) | Swin-T | 74.0 | 91.7 | 81.5 | 47.8 | 74.7 |
| | *UniPose* (Image) | Swin-T | 73.6 | 91.9 | 80.6 | 47.2 | 74.2 |
| | *UniPose* (Text) | Swin-L | 79.2 | 95.7 | 87.2 | 58.3 | 79.8 |
| | *UniPose* (Image) | Swin-L | 79.0 | 95.7 | 86.8 | 57.0 | 79.6 |

HRNet: AP-10K, 58K training data
ED-Pose: COCO + Human-Art + AP-10K + APT-36K, 154K training data
ViTPose++: COCO+ COCO-W + MPII + AIC + AP-10K + APT-36K, 387K training data
*UniPose*: *UniKPT*, 226K training data

Table 16: Comparisons with the state-of-the-art open-vocabulary object detector, focusing on instance-level and keypoint-level detection. ‡ denotes the fine-tuning of GroundingDINO using the keypoint detection datasets. Notably, we limit the instance-level comparison to $AP_M$ (medium objects) and $AP_L$ (large objects), as small objects do not have keypoints annotated.

| Methods | Backbone | Instance-level $AP_M$ | $AP_L$ | Keypoint-level AP | $AP_M$ | $AP_L$ | Training Datasets | Dataset Volume |
|---|---|---|---|---|---|---|---|---|
| *COCO* `val set` | | | | | | | | |
| GroundingDINO (Text) | Swin-T | 70.8 | 82.0 | 3.1 | 2.8 | 3.2 | O365,GoldG,Cap4M | 1858K |
| GroundingDINO (Text) | Swin-B | 69.7 | 79.5 | 6.8 | 6.6 | 7.2 | COCO,O365,GoldG,Cap4M,OpenImage,ODinW-35,RefCOCO | - |
| GroundingDINO‡ (Text) | Swin-T | 71.2 | 83.4 | 1.8 | 1.7 | 1.9 | COCO,Human-Art,AP-10K,APT-36K | 1858K + 155K |
| *UniPose* (Text) | Swin-T | 71.1 | 80.2 | **74.2** | **68.8** | **82.1** | COCO,Human-Art,AP-10K,APT-36K | 155K |
| *UniPose* (Image) | Swin-T | 71.1 | 80.3 | 74.1 | **68.8** | 81.8 | COCO,Human-Art,AP-10K,APT-36K | 155K |
| *Human-Art* `val set` | | | | | | | | |
| GroundingDINO (Text) | Swin-T | 11.5 | 27.0 | 2.1 | 1.7 | 2.3 | O365,GoldG,Cap4M | 1858K |
| GroundingDINO (Text) | Swin-B | 13.3 | 27.9 | 4.4 | 3.7 | 4.5 | COCO,O365,GoldG,Cap4M,OpenImage,ODinW-35,RefCOCO | - |
| GroundingDINO‡ (Text) | Swin-T | 33.3 | **67.0** | 1.4 | 0.8 | 1.5 | COCO,Human-Art,AP-10K,APT-36K | 1858K + 155K |
| *UniPose* (Text) | Swin-T | 33.7 | 63.1 | **72.2** | **39.5** | **76.7** | COCO,Human-Art,AP-10K,APT-36K | 155K |
| *UniPose* (Image) | Swin-T | **34.0** | 63.0 | 71.8 | 39.3 | 76.4 | COCO,Human-Art,AP-10K,APT-36K | 155K |
| *AP-10K* `val set` | | | | | | | | |
| GroundingDINO (Text) | Swin-T | 5.1 | 13.7 | 1.3 | 0.6 | 1.3 | O365,GoldG,Cap4M | 1858K |
| GroundingDINO (Text) | Swin-B | 29.1 | 44.1 | 7.8 | 5.4 | 8.2 | COCO,O365,GoldG,Cap4M,OpenImage,ODinW-35,RefCOCO | - |
| GroundingDINO‡ (Text) | Swin-T | **56.5** | **79.7** | 0.7 | 0.4 | 1.0 | COCO,Human-Art,AP-10K,APT-36K | 1858K + 155K |
| *UniPose* (Text) | Swin-T | 54.5 | 78.8 | **73.2** | 45.6 | **74.3** | COCO,Human-Art,AP-10K,APT-36K | 155K |
| *UniPose* (Image) | Swin-T | 55.8 | 79.0 | 72.8 | **47.2** | 74.0 | COCO,Human-Art,AP-10K,APT-36K | 155K |

define textually and visually, especially when only a single source image is inputted. These objects also tend to have similar local features in their visual appearance and global structural features that are not sufficiently distinct. Third, Although our method shows superiority in multi-object detection scenarios, we find that recognition in places of extreme occlusion or heavily invisible keypoints, which may lose either local or global visual structure information, is still challenging (see Fig. 8(c)).

# D BROADER IMPACT AND LIMITATION

**Broader Impact:** Based on the proposed *UniPose*, we can provide 1) a keypoint generalist for any category and keypoint, including articulated (e.g., human and animal), rigid (e.g., car and chair), and soft (e.g., cloth and dress) objects, which has great potential to benefit various downstream areas, such as robot automation, human-object interaction, and AR/VR; 2) a better fine-grained text to local region alignment with structure and direction knowledge that can provide a text-region similarity score for fine-grained visual perception and vision-language understanding; That is, the similarity score can be used for evaluation or interpretability; 3) a user-friendly connector with either natural language or visual prompts to first detect keypoints and then take them as user clicks for fine-grained detection, segmentation, and tracking. To sum up, we hope this work could broaden the way to fine-grained open-vocabulary and category-agnostic perception tasks from a keypoint representation.

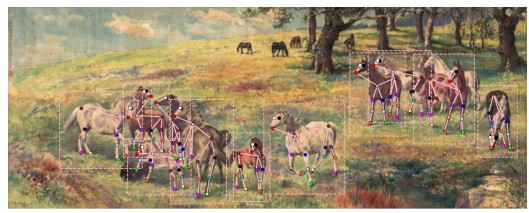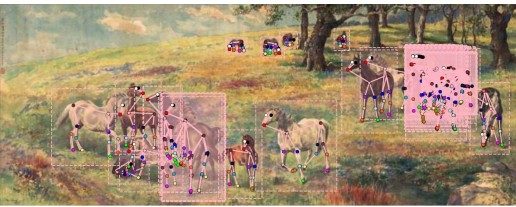

(a) Imbalance threshold between precision and recall

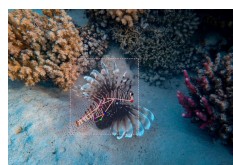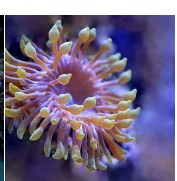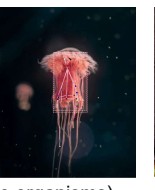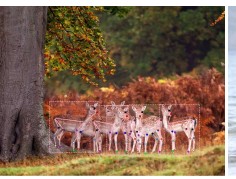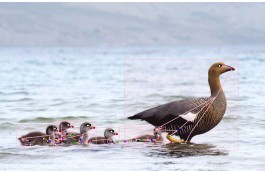

(b) Complex and flexible objects (e.g., marine organisms)  (c) Heavy occlusions and invisibility

Figure 8: We show three types of failure cases in our method.

**Limitation:** In addition to the limitations described in this section C.3, we believe the main issue currently lies in the data. Although this work has aggregated thirteen keypoint datasets, there are three limitations. First, compared with the amount of training data for CLIP, we still have 1,000 times less data, leaving room for improvement in the performance and generalization of the model. Second, some super-species with novel topologies that are not included (e.g., Kingdom Fungi and Kingdom Plantae) make the model hard to generalize to these cases. Lastly, we did not consider fine-grained detection and segmentation datasets (Shao et al., 2019; He et al., 2022a) to strengthen the unified datasets. All the above issues will be left as future work.

