# OpenReview forum: "UniPose: Detecting Any Keypoints"
_ICLR.cc/2024/Conference — ICLR 2024 Conference Withdrawn Submission_

### Official Review · Reviewer_Di8P · 2023-10-25

**Soundness:** 2 fair
**Presentation:** 1 poor
**Contribution:** 3 good
**Rating:** 3
**Confidence:** 4

**Summary:**

The authors proposed a unified framework named UniPose to estimate keypoints for any object with visual and tuxtual prompts. The trained model could generalize to cross-instance and cross-keypoint classes, as well as various styles, scales, and poses.  The authors unified 13 keypoint detection datasets with 338 keypoints across 1,237 categories over 400K instances, and employed such large-scale dataset to train the UniPose model, and obtained a generalist keypoint detector.

**Strengths:**

1) This work arranges a large scale keypoint detection dataset.
2) The paper proposes a large keypoint detection model with visual and tuxtual prompts.
3) The proposed model shows good performance in generalization.
4) The experiments are numerous.

**Weaknesses:**

Overall, the current version of paper does not provide sufficient explanations and in-depth analyses on major points. There are lots of experiments on comparison, but the in-depth experimental analyses (e.g., on prompts) are more necessary and meaningful.

- The most important one weakness is that there is no clear explanation for the textual prompts of keypoints. For examples, clothing or table could have ambiguous keypoint names, how to employ textual prompts to detect their keypoints?

- Another important issue is that all the visualizations are shown without prompts and thus difficult to analyse. It is critical for understanding the effects of various prompts.

- In Section 4.1, why not apply UniPose in the setting of class-agnostic pose estimation as in CapeFormer? The comparison with adapted CapeFormer is unfair and unclear for indicating the performance of UniPose on detecting unseen objects.

- Lacking analysis for Section 4.3. The performance gaps are trivial, considering the differences on data and tasks.

- In Section 4.4, how to evaluate the SOTA open-vocabulary object detector on keypoint-level detection? Is is feasible to represent each keypoint with a small bbox in fine-tuning?

- Why does CapeFormer drop a lot in Tab. 12? Are the training and test resolutions of CapeFormer consistent?

- What are UniPose-T and UniPose-V in Tab. 4?

- In Section 4.4, what is CLIP score? How to analyse the CLIP scores of UniPose and CLIP in Fig. 7?

**Questions:**

See Weaknesses*

---

> ### Author Response · Authors · 2023-11-15
> **Rebuttal by Authors (Part 1)**
>
> **1. The most important one weakness is that there is no clear explanation for the textual prompts of keypoints. For examples, clothing or table could have ambiguous keypoint names, how to employ textual prompts to detect their keypoints?**
>
> During the construction of our UniKPT dataset, one of the great efforts is to address the issues of ambiguous keypoint names in existing datasets. We merge and unify all the keypoints from 13 datasets into 338 keypoints in the UniKPT dataset. And we take great care in annotating each keypoint with comprehensive textual descriptions, incorporating rich directional information such as 'upper,' 'lower,' 'left,' and 'right,' and assigning each keypoint a unique name to add the fine-grained semantic meaning and prevent any conflicts. We will provide a comprehensive list of 338 textual prompts for keypoints in the Appendix of the revised version.
>
> For instance, in the DeepFashion2 dataset, keypoints are identified solely by numerical labels (e.g., 1, 2, 3, 4) for clothing items. In contrast, our annotation process involves the detailed labeling of specific keypoints. For example, when annotating the neckline, we distinguish between six distinct keypoints: 'upper center neckline,' 'upper right neckline,' 'lower right neckline,' 'lower center neckline,' 'lower left neckline,' and 'upper left neckline.' This enables our model to achieve fine-grained text-to-keypoint alignment and utilize corresponding keypoint prompts for precise detection.
>
> **2. Another important issue is that all the visualizations are shown without prompts and thus difficult to analyse. It is critical for understanding the effects of various prompts.**
>
> Thanks for the suggestions. We will supplement the corresponding prompts along with visualizations in the revised version.
> For in-the-wild images presented in Figure 2, we employ textual prompts during testing, where these textual prompts are user-friendly and are particularly suitable for large-scale image testing.
> We default to using the keypoints description defined by AP-10K for animal categories. For the human category, we opt for the keypoints description defined by COCO. The results show that UniPose demonstrates exceptional fine-grained localization capabilities and its ability to generalize effectively across different image styles, instances, and poses.
>
>
> **3. In Section 4.1, why not apply UniPose in the setting of class-agnostic pose estimation as in CapeFormer? The comparison with adapted CapeFormer is unfair and unclear for indicating the performance of UniPose on detecting unseen objects.**
>
> In Table. 2, we indeed have conducted a fair comparison with CapeFormer and existing methods in the same class-agnostic pose estimation setting using the MP-100 dataset for training and testing. CapeFormer is a top-down-based method that utilizes ground-truth bounding boxes to crop images into individual object images for performing class-agnostic pose estimation. However, UniPose is an end-to-end method with a significant advantage in its ability to generalize to different objects and detect new ones. To maintain consistency with CapeFormer's settings, we also use ground-truth bounding boxes, foregoing object generalization. The results show that UniPose, as an end-to-end framework, could surpass existing top-down-based approaches with a superior inference time.
>
> Furthermore, it is essential to highlight UniPose's competence in conducting detection and pose estimation in multi-object, multi-class scenarios—an achievement CapeFormer cannot attain. As demonstrated in Table. 3 and 12, we present results without ground-truth bounding boxes and test CapeFormer in single-/multi-object scenarios. These results underscore a notable performance decrease for CapeFormer in single-object scenarios (where disguising the background and object is challenging), and it demonstrates an inability to effectively handle scenes with multiple objects.
>
>
> **4. Lacking analysis for Section 4.3. The performance gaps are trivial, considering the differences on data and tasks.**
>
> Existing general models focus on training models to handle multiple vision tasks. However, in the field of pose estimation, their performance is not optimal due to the gap involved in handling multiple tasks, even when they employ large models. Also, due to the limited scope for pose estimation, they all tend to focus solely on tasks like person keypoint detection (e.g., COCO) or animal keypoint detection (e.g., AP-10K).
>
> In comparison, our focus is to unify the keypoint detection tasks and target a keypoint generalist that achieves effectiveness and generality in keypoint detection across any articulated (e.g., human, animal, and detailed face and fingers), rigid (e.g., vehicle), and soft objects (e.g., clothing).
> In section 4.3, we compare UniPose with existing general models to demonstrate the effectiveness of UniPose as a powerful generalist model in the field of pose estimation.

---

> > ### Author Response · Authors · 2023-11-15
> > **Rebuttal by Authors (Part 2)**
> >
> > **5. In Section 4.4, how to evaluate the SOTA open-vocabulary object detector on keypoint-level detection? Is is feasible to represent each keypoint with a small bbox in fine-tuning?**
> >
> > Multi-object keypoint detection task need to detect both the bounding boxes of individual objects and their corresponding keypoints. Directly using the keypoint prompt in Grounding-DINO only outputs a set of keypoints without knowing which instance these keypoints belong to. Thus, we conduct two steps for evaluation: First, we utilize the instance text prompt to detect all candidate instance boxes. Next, we employ these instance boxes to crop the images into individual single-object images. Subsequently, we use the keypoint prompt to perform keypoint detection on these single-object images, wherein we select the top-1 score's box center as the keypoint position.
> >
> > In practice, directly transforming each keypoint with a small box for fine-tuning Grounding-DINO leads to learning only part-level localization. However, as mentioned above, the task of pose estimation is distinct from this, as it requires the identification of instances and the matching of a set of keypoints. This distinction underscores the technical contribution of our work, as we introduce the first open-vocabulary end-to-end keypoint detector that can generalize across objects and keypoints simultaneously.
> >
> > **6. Why does CapeFormer drop a lot in Tab. 12? Are the training and test resolutions of CapeFormer consistent?**
> >
> > CapeFormer solely detects keypoint on the single-object images.
> > In Table 2, the performance drop demonstrates two key observations: (1) CapeFormer struggles to handle multi-object scenes when we change the cropped single-object image into the original multi-object image, and (2) CapeFormer fails to handle larger test image resolution while previous pose estimation methods, including UniPose, have demonstrated improved performance when the testing resolution exceeds the training resolution. CapeFormer has primarily learned keypoint matching at a specific resolution, making it lack robustness to accommodate scale variations, even in single-object scenarios.
> >
> > **7. What are UniPose-T and UniPose-V in Tab. 4?**
> >
> > As in the caption of Table. 5, T and V denote textual and visual prompts used for inference, respectively. We will elaborate on them in the revised version.
> >
> >
> >
> > **8. In Section 4.4, what is CLIP score? How to analyse the CLIP scores of UniPose and CLIP in Fig. 7?**
> >
> > The CLIP score is a commonly used metric designed to quantify the similarity between text and images. Specifically, both UniPose and CLIP take the test image and textual prompt as input, generating image features and text features as outputs. The CLIP score measures the similarity between these two features.
> > The original CLIP primarily focuses on image-level text-to-image alignment. In contrast, UniPose performs either text-to-instance or text-to-keypoint alignment, enhancing global and local cross-modality capability (e.g., localizing the cat's right eye).

---

### Official Review · Reviewer_Ldy8 · 2023-10-30

**Soundness:** 3 good
**Presentation:** 3 good
**Contribution:** 3 good
**Rating:** 6
**Confidence:** 3

**Summary:**

This paper makes the first attempt to propose an end-to-end coarse-to-fine keypoints detection framework named UniPose trained on a unified keypoint dataset, which can detect the keypoints of any object from the instance to the keypoint levels via either visual prompts or textual prompts and has remarkable generalization capabilities for unseen objects and keypoints detection. This work unifies 13 keypoint detection datasets containing 338 keypoints detection over 400K instances to train a generic keypoint detection model. Compared to the state-of-the-art CAPE method, this method exhibits a notable 42.8% improvement in PCK performance and far outperforms CLIP in discerning various image styles.

**Strengths:**

* Originality：
The first attempt of an end-to-end keypoints detection framework is developed by combining visual and textual prompts. Through the mutual enhancement of textual and visual prompts, this method can have strong fine-grained localization and generalization abilities for class-agnostic pose estimation and multi-class keypoints detection tasks.

* Quality：
The method description is detailed, and it is illustrated in the accompanying figures.

The experiment is fully configured, taking into consideration unseen objects, expert model, general model, and open-vocabulary model. Comparative tests are conducted.

* Clarity
The paper exhibits a well-structured logical flow, accompanied by an appendix that provides an extensive overview of the work, including the introduction of the dataset, supplementary experiments, algorithmic limitations, and more.

* Significance
This paper Unifies 13 datasets to build a unified keypoints detection dataset named UniKPT. And the authors say that each keypoint has its own text prompts, and each category has its default set of structured key points. This unified dataset with visual and textual prompts can provide data support for point detection tasks in future work.

**Weaknesses:**

There are no complete examples of textual prompts.

In 4.4, the Fig.7 should be Tab.7.

**Questions:**

For the Inference Pipeline, whether only one prompt is used as input, and whether it will work better if two prompts are used?

---

### Official Review · Reviewer_SvA3 · 2023-10-31

**Soundness:** 2 fair
**Presentation:** 2 fair
**Contribution:** 2 fair
**Rating:** 5
**Confidence:** 5

**Summary:**

This paper proposes a so-called UniPose to detect various types of keypoints. The UniPose takes text or image with keypoint annotations as prompts, in order to detect the corresponding keypoints in query image. In order to support both modalities of prompts, the visual prompt encoder, textual prompt encoder, and two decoders are developped. The model is trained on 13 keypoint detection datasets with 338 keypoints, and the results show it can detect varying types of keypoints to some extent.

**Strengths:**

1) A model which supports textual or visual prompts are proposed.

2) The visualization shows the effectiveness to some extent.

**Weaknesses:**

1) Reading through the paper, it lets reviewer feel that the writing requires significant improvement and the math symbols are in mess. Moreover, there are already many works in literature using visual prompts such few-shot keypoint detection [1,2,3,4], and also the works regarding textual prompts such as [5,6]. The paper should fully discuss these two types of related works. From the technical perspective, it shows little advance compared to existing works. The simple combination of both types of prompts make the contribution and novelty of this work weak.

    [1] Metacloth: Learning unseen tasks of dense fashion landmark detection from a few samples @ TIP21

    [2] Few-shot keypoint detection with uncertainty learning for unseen species @ CVPR'22

    [3] Pose for everything+Towards category-agnostic pose estimation @ ECCV'22

    [4] Few-shot Geometry-Aware Keypoint Localization @ CVPR'23

    [5] Clamp: Prompt-based contrastive learning for connecting language and animal pose @ CVPR'23

    [6] Language-driven Open-Vocabulary Keypoint Detection for Animal Body and Face @ arxiv'23

2) Some claims may not be true. For example, ``Xu et al. (2022a) first proposed the task of category-agnostic...". Work [3] may not be the first work as [1-2] are earlier. Moreover, what is the meaning of "the keypoint to keypoint matching schemes without instance-to-instance matching are not effective"? The keypoint representation can also aggregate the global information or context.

3) Given the existence of a similar approach such as CLAMP [5], it's essential to evaluate the performance of your method when compared to CLAMP. This evaluation should be conducted on the Animal pose dataset within the context of a five-leave-one-out setup. In this setting, the model is trained on four different species while being tested on the remaining one species. The paper should include the results of PCK@0.1 to facilitate meaningful comparisons.

3) Coarse-to-fine strategy already appears in paper [2023-ICLR-Explicit box detection unifies end-to-end multi-person pose estimation]; while the ideas of using prompts based keypoint detection already appears in FSKD, CLAMP, etc.

4) Some details are missing. This work uses CLIP as image and text encoder. The CLIP generally takes an image with size of 224 as input, while in table 2 and 3, the UniPose takes original image or image with size of 800 as input. Will the high-resolution input slow down the speed? What is the size of feature map after passing image (e.g. 800) through CLIP? A step forward, will the CLIP retain its prior knowledge after using such a high-resolution input?

    Moreover, how to select the visual object feature and textual object feature to produce $Q_{obj}$? How to select if both exist?

5) In table 1, some of datasets are already included in MP-100. For example, COCO, AP-10K, etc. So what is the meaning of counting it again to build the dataset of UniKPT?

**Questions:**

See weaknesses.

**Details Of Ethics Concerns:**

.

---

> ### Author Response · Authors · 2023-11-15
> **Rebuttal by Authors (Part 1)**
>
> **1. Reading through the paper, it lets reviewer feel that the writing requires significant improvement and the math symbols are in mess.**
>
> Thanks for the suggestions. Our main article has aligned mathematical symbols with our implementation details. We will carefully refine this version and improve the notation in the revised edition. If you have specific examples in mind, we would be grateful if you could share them, and we can focus on those for better revisions.
>
> **2. There are already many works in literature using visual prompts such few-shot keypoint detection [1,2,3,4], and also the works regarding textual prompts such as [5,6]. The paper should fully discuss these two types of related works. From the technical perspective, it shows little advance compared to existing works. The simple combination of both types of prompts make the contribution and novelty of this work weak.**
>
>
> First and foremost, we would like to emphasize that UniPose is the first end-to-end framework that can generalize to unseen objects and keypoints in multi-object, multi-class scenes. In comparison, **all the methods listed above [1-6] are top-down approaches, as they rely on ground-truth bounding boxes to crop a multi-object image into several single-object images and subsequently employ visual or textual prompts for the following single-object keypoint detection.** Thus, these methods are unable to address multi-class multi-object scenarios without known instance-level object detection, particularly situations where an image contains numerous objects of different categories with varying keypoint definitions.
>
>
> Secondly, UniPose is the first multi-modal prompt-based pose estimator. However, **the methods [1-6] only support a single modal prompt**. Only supporting visual prompts makes the user interaction unfriendly and inefficient, while only supporting textual prompts lack fine-grained low-level visual information and make it hard to localize the indescribable positions. UniPose jointly leverages visual and textual prompts for training via cross-modality contrastive learning to boost the generalization and effectiveness of any keypoint detection. We have demonstrated the mutual benefit between the two kinds of prompts in Table. 10.
>
>
>
> Thirdly, UniPose targets a keypoint generalist that achieves effectiveness and generality in keypoint detection across any articulated (e.g., human, animal, and detailed face and fingers), rigid (e.g., vehicle), and soft objects (e.g., clothing), which is trained on the proposed UniKPT dataset.
> In contrast, references [1] or [2] primarily focus on single-object keypoint detection, specifically within the clothing or animal super-categories. Reference [3,4], despite handling more objects, exhibits limitations in terms of generalizability and effectiveness due to relatively small-scale training data.
> As for reference [5], as detailed in our related work section, CLAMP concentrates solely on single-object animal keypoint detection. It leverages CLIP with language guidance to prompt animal keypoints containing a fixed keypoint set (e.g., 20) and doesn't consider keypoint-level open vocabulary. The primary emphasis of CLAMP lies in cross-species generalization within a predefined skeleton structure, a domain where UniPose has made substantial advancements.
> Regarding reference [6], it's noteworthy that this work was submitted to Arxiv in October 2023, coinciding with our work. Also, it focuses on single objects, with a specific focus on animal body and facial parts.
>
> **3. Some claims may not be true. For example, ``Xu et al. (2022a) first proposed the task of category-agnostic...". Work [3] may not be the first work as [1-2] are earlier.**
>
> In fact, we follow the definition of the category-agnostic pose estimation (CAPE) task by [3]. This task requires the pose estimator to detect keypoints of arbitrary categories, such as animals, clothing, furniture, and persons, given the keypoint definitions.
>
> [1] and [2] primarily concentrate on a single super category like clothing or animals. We will clarify this distinction and explain it in the revised version.

---

> > ### Author Response · Authors · 2023-11-15
> > **Rebuttal by Authors (Part 2)**
> >
> > **4. What is the meaning of "the keypoint to keypoint matching schemes without instance-to-instance matching are not effective"? The keypoint representation can also aggregate the global information or context.**
> >
> > This clarification is based on previous category-agnostic pose estimation (CAPE) task methods. Existing CAPE works firstly use the ground-truth box to crop the multi-object images into single-object images. That means if there is no effective open-vocabulary object detector, these methods will fail at the first step. Then, they employ keypoint visual prompts for keypoint-to-keypoint matching within the context of a single object. However, it's important to recognize that this scheme may not effectively generalize to real-world scenarios where images often contain multiple unseen objects and lack available ground-truth bounding boxes.
> >
> > In contrast, UniPose is an end-to-end framework that incorporates instance-to-instance matching, allowing it to capture high-level semantic and object-level relationships. This enables UniPose to learn open-vocabulary object detection, enhancing fine-grained keypoint detection for unknown classes, and avoiding a sole focus on low-level local appearance transformations.
> >
> > **5. Given the existence of a similar approach such as CLAMP [5], it's essential to evaluate the performance of your method when compared to CLAMP. This evaluation should be conducted on the Animal pose dataset within the context of a five-leave-one-out setup. In this setting, the model is trained on four different species while being tested on the remaining one species. The paper should include the results of PCK@0.1 to facilitate meaningful comparisons.**
> >
> > In terms of methodology, UniPose's similarity with CLAMP lies in the utilization of the CLIP text encoder to encode the keypoint prompts. However, this operation is common in language-guided methods. It's important to highlight that CLAMP's focus is not on handling open-vocabulary keypoints but centers around achieving cross-species generalization under a fixed skeleton. Additionally, CLAMP cannot handle multi-object, multi-class scenes as it only targets single-object scenarios.
> >
> >
> > The five-leave-one-out setting in the AnimalPose dataset is used to demonstrate cross-species generalization, specifically focusing on mammals within a fixed skeleton. However, as shown in Figures 1 and 2, UniPose has exhibited remarkable capabilities by extending its generalization to previously unencountered species, including insects.
> >
> > More importantly, we want to emphasize that UniPose exhibits strong generalization across unseen objects and keypoints. To underscore this, we present the zero-shot performance of UniPose on AnimalPose, a dataset not included in UniPose's training data. We compare this zero-shot performance to that of CLAMP, which is specifically trained on AnimalPose. The results demonstrate that our zero-shot performance closely approaches that of the CLAMP model trained on AnimalPose, highlighting our robust capabilities in object and keypoint generalization.
> >
> >
> > | Methods | Backbone  | ${\rm AP}$ | ${\rm AP}_{M}$ | ${\rm AP}_{L}$ |
> > |---------|-----------|------------|----------------|----------------|
> > | CLAMP   | ResNet-50 | 72.5       | 67.9           | 73.8           |
> > | CLAMP   | ViT-Base  | 74.3       | 71.9           | 75.2           |
> > | UniPose | Swin-T    | 62.5       | 59.8           | 67.4           |
> > | UniPose | Swin-L    | 70.1       | 67.0           | 73.2           |
> >
> > Table. Evaluation of UniPose's zero-shot performance on the AnimalPose dataset. **The CLAMP model is trained on the AnimalPose dataset and tested on the same dataset.** All the methods use the grounding-truth box to ignore the instance detection for a fair comparison.
> >
> > In the revised version, we will include this experiment and provide supplementary PCK@0.1 results on the corresponding datasets that are measured using the PCK metric.

---

> > > ### Author Response · Authors · 2023-11-15
> > > **Rebuttal by Authors (Part 3)**
> > >
> > > **6. Coarse-to-fine strategy already appears in paper [2023-ICLR-Explicit box detection unifies end-to-end multi-person pose estimation]; while the ideas of using prompts based keypoint detection already appears in FSKD, CLAMP, etc.**
> > >
> > > The objective of this work is to introduce an end-to-end prompt-based keypoint generalist that is capable of generalizing across various objects, image styles, categories, and poses on multi-object keypoint detection.
> > >
> > > Firstly, we have admitted that UniPose is based on the DETR-like end-to-end non-promptable
> > > human pose estimator (ED-Pose) in the main paper. Both works can first decode the instance information and then decode the corresponding fine-grained keypoints to provide a coarse-to-fine information flow. However, ED-Pose is a non-promptable multi-person pose estimator that focuses on single-class keypoint detection, making it hard to generalize across various categories. As in Table 4, we compare UniPose to ED-Pose under a fair multi-dataset training setting. UniPose outperforms ED-Pose across all datasets in terms of both instance-level and keypoint-level detection. Moreover, for the AP-10K dataset, which involves the classification of 54 different species, UniPose surpasses ED-Pose with a 27.7 AP improvement. These improvements are owed to our proposed multiple prompt-based strategies with instance-level and keypoint-level alignments.
> > >
> > > Secondly, both FSKD and CLAMP support only a single modal prompt and are primarily focused on single-object keypoint detection. In contrast, UniPose takes advantage of the ability to jointly utilize visual and textual prompts during training through cross-modality contrastive learning with instance-level and keypoint-level alignments. Table 10 illustrates the mutual benefits derived from employing both types of prompts.
> > >
> > > **7. Some details are missing. This work uses CLIP as image and text encoder. The CLIP generally takes an image with size of 224 as input, while in table 2 and 3, the UniPose takes original image or image with size of 800 as input. Will the high-resolution input slow down the speed? What is the size of feature map after passing image (e.g. 800) through CLIP? A step forward, will the CLIP retain its prior knowledge after using such a high-resolution input?**
> > >
> > >
> > > As shown in Figure. 2 and 5(a), the input to the CLIP image encoder is the visual prompt image. It only contains the single object and the corresponding keypoint definitions. The resolution size is set at 224, aligning it with the requirements of CLIP.
> > > The resolution of the target input image (including multiple objects) is 800. It aligns with the typical input resolution used in other end-to-end models like ED-Pose.
> > >
> > > **8. How to select the visual object feature and textual object feature to produce
> > > ? How to select if both exist?**
> > >
> > >
> > > During training, we employ a 50\% probability to randomly select either a visual prompt or a textual prompt for each iteration. In the inference phase, users have the flexibility to input any of the available prompts as desired. From our quantitative results, we report the results of both prompts as the inputs. For the visualization, we simply choose the textual prompts (user-friendly) to test in-the-wild images.
> > >
> > > **9. In table 1, some of datasets are already included in MP-100. For example, COCO, AP-10K, etc. So what is the meaning of counting it again to build the dataset of UniKPT?**
> > >
> > > UniKPT has a clear significance from the number of images/instances/categories over 10 times than MP-100 and the quality of textual labels. This extensive dataset will benefit large-scale training and serve the research community.
> > > As described in the main paper (the key motivation of the proposed UniKPT), the MP-100 dataset comprises only 20,000 images, featuring 100 instance classes extracted from existing datasets (e.g., using 200 images in AP-10K). In contrast, from Table 1, UniKPT unifies a wide range of existing datasets to obtain 226,000 images with 1237 instance classes.
> > >
> > > More importantly, we merge and unify all the keypoints from 13 datasets into 338 keypoints. We took great care in annotating each keypoint with comprehensive textual descriptions, incorporating rich directional information such as 'upper,' 'lower,' 'left,' and 'right,' and assigning each keypoint a unique name to add the fine-grained semantic meaning and prevent any conflicts.

---

### Official Review · Reviewer_WBby · 2023-11-07

**Soundness:** 3 good
**Presentation:** 4 excellent
**Contribution:** 3 good
**Rating:** 6
**Confidence:** 3

**Summary:**

The paper discusses a new framework called UniPose, which aims to detect keypoints in various objects, including articulated, rigid, and soft ones, using visual or textual prompts. Keypoints are pixel-level representations of objects, especially articulated ones. Current fine-grained promptable tasks focus on object instance detection and segmentation but struggle with identifying detailed structured information, such as eyes, legs, or paws. The UniPose framework is the first attempt to create an end-to-end prompt-based keypoint detection system that can be applied to any object. It unifies various keypoint detection tasks and leverages multiple datasets to train a generic keypoint detection model. UniPose aligns textual and visual prompts through cross-modality contrastive learning optimization, resulting in strong fine-grained localization and generalization capabilities across different image styles, categories, and poses. The framework is expected to enhance fine-grained visual perception, understanding, and generation.

**Strengths:**

1. This paper is well-written and easy to follow.
2. This article provides a comprehensive summary of previous keypoint research, highlighting both its strengths and weaknesses, and uses this as motivation to propose its own approach.
3. The methodology in this article is well-designed, combining both text and image prompts for keypoint detection .

**Weaknesses:**

This article combines two approaches to prompts, but lacks in-depth analysis of the strengths and weaknesses of both modalities for this task:

1.What are the advantages and disadvantages of each prompt individually, and can you provide some visual results?
2. Can the strengths and weaknesses of the two prompts complement each other?
3. Is it possible to dynamically weight the two prompts? For example, can text be prioritized when there are no suitable images to serve as prompts?

**Questions:**

Please see the weakness. If you address my concerns, I am willing to improve my score. Thanks.